# Static Analysis of Mobile Pump Truck Frame for Four Typical Working Conditions

**San-Ping Li [1], Hai-Bin Lin [1], Yu-Liang Zhang [2],* and Liang Cheng [2]**

1   College of Mechanical and Electrical Engineering, Northeast Forestry University, Harbin 150040, China; bluelii73@163.com (S.-P.L.); linhaibin@nefu.edu.cn (H.-B.L.)
2   Key Laboratory of Air-Driven Equipment Technology of Zhejiang Province, College of Mechanical Engineering, Quzhou University, Quzhou 324000, China; chengliang@qzc.edu.cn
*   Correspondence: zhang002@sina.com

**Abstract:** In order to verify the safety and reliability of the mobile pump truck, this paper takes the frame of a certain type of mobile pump truck as the research object. Through the establishment of a finite element model, four kinds of materials including Q345 and other types of steel are used to define the body parts, the four typical motion situations of the mobile pump truck are analyzed statically, and the maximum stress and deformation of the mobile pump truck under four working conditions are obtained. The results show that the stress and deformation generated by the mobile pump truck under full load bending, emergency turning, and emergency braking conditions are relatively small; while they generate significant stress and deformation under torsional conditions, they all meet the strength design requirements. Among them, the maximum stress and maximum displacement under the full load bending condition are 71.76 MPa and 2.11 mm; the maximum stress and maximum displacement under the full load torsion condition are 352.68 MPa and 18.18 mm; the maximum stress and maximum displacement under emergency turning conditions are 79.718 MPa and 2.68 mm; and the maximum stress and maximum displacement under emergency braking conditions are 74.907 MPa and 2.81 mm. The analysis results can provide a reference basis for the design of the mobile pump truck frame in the future.

**Keywords:** mobile pump truck; workbench; finite element analysis; statics analysis

## 1. Introduction

In recent years, the mobile pump truck has been widely used for flood control, drainage, crop irrigation, drought resistance, and temporary water pumping in areas without electricity, as well as field water supply and island freshwater collection. As the main bearing component of the mobile pump truck, the frame is easy to be damaged due to insufficient strength and stiffness in the process of operation and driving. Therefore, it is very important to study the strength of the frame of the mobile pump truck to improve the safety and reliability of the whole mobile pump truck.

Ilham Widiyanto et al. carried out finite element analysis on the static loading of the automobile chassis model, and compared the materials used from five aspects: stress, strain, displacement, reaction force, and safety factor, so as to seek out the best automobile chassis material [1]. Girish Dutt Gautam et al. used a finite element method to analyze the tubular steel anti-roll frame of a formula racing car, and studied and discussed the influence of different loading conditions on the maximum bearing capacity and maneuvering performance of structural members [2]. P. Satheesh Kumar Reddy et al. used finite element analysis software to analyze the static, free vibration and tension buckling of the drive shaft [3]. Marco Cavazzuti et al. combined topology, size optimization, and finite element analysis to study the structural performance of an automobile chassis [4]. Li et al. established the finite element model of the body-in-white,

calculated the bending stiffness and torsional rigidity of the body according to the actual working conditions of the vehicle, and studied the main opening deformation of the body under bending and torsional conditions [5]. Chen et al. studied the working conditions and load characteristics of the subframe of the YJ3128 dump truck, and analyzed the stress of the subframe using Ansys. According to different stress states, the causes of fatigue cracks were studied [6]. Meng et al. used the software HyperWorks to analyze and study the static stiffness and modal of a machine bed, optimized the design on this basis, and obtained the lightest structural shape of the bed [7]. Chen et al. used Ansys Workbench software to carry out modal analysis, topology optimization, and finite element analysis of the bending stiffness and torsional stiffness of an electric commercial vehicle frame model, providing a basis for structure optimization and a lightweight vehicle [8]. Andrzej Banaszek et al. studied the impact of corrosion on the safety of hydraulic pipelines installed on product tankers and chemical tankers, and analyzed the impact of erosion or corrosion on the failure rate of load-bearing structures [9]. Ou et al. carried out crashworthiness analysis, mode frequency, and dynamic stiffness analysis of key attachment points, as well as BIW stiffness and component connection stiffness analysis on the model under a full load [10]. Javad Gholami et al. used the finite element method to calculate and analyze the excavator bracket, providing a basis for the subsequent optimization design [11]. Andrzej Banaszek et al. used the finite element method to study the stress effect of installation methods on the main lines on chemical storage tanks [12]. Usama Idrees et al. used Ansys to simulate cars under different speeds and obstacles, and analyzed the impact of vehicle collision speed on the passenger area [13]. Gabriel Nagy used the finite element method to analyze the stress of the tank car body and compared it with the experiment, thus predicting the potential danger area of the tank car body [14]. Tomasz Urbaumski et al. conducted relevant research on the deformation of fixed plate edges due to butt joints, and analyzed the technical and structural parameters to evaluate the deformation shape [15]. Kirthana et al. used the finite element method to optimize the topology of the engine mounting bracket. By studying different material layouts and different designs, the optimal model was obtained via calculation, analysis, and the comparison of stress and weight [16]. Hu et al. used numerical simulation technology to study the key response parameters of passenger cars driving on six different road surfaces, so as to study the durability of passenger cars [17].

From the above, it can be seen that research on the frame structure at home and abroad mainly focuses on ordinary passenger vehicles, while research on the frame structure of mobile pump trucks that play an important role in emergency rescue has almost not been involved. Based on this, this paper carries out statics analysis on the frame of a mobile pump truck to verify the safety and reliability of the mobile pump truck when it is working. The analysis results can provide a reference for the subsequent design of the mobile pump truck frame.

## 2. Frame Model and Calculation Method

### 2.1. Frame Model

The length of the 3D model of the frame was 3342 mm, the width was 1700 mm, and the height was 734 mm. In the simulation process, the model was established according to the 1:1 ratio. Considering the main factors, simplify the process holes on the non-stressed parts, crossbeams, and longitudinal beams in the body structure, and simplify the chamfers and rounded corners, which have little impact on the strength of the car body, so as to obtain the simplified frame model. The simplified model is shown in Figure 1.

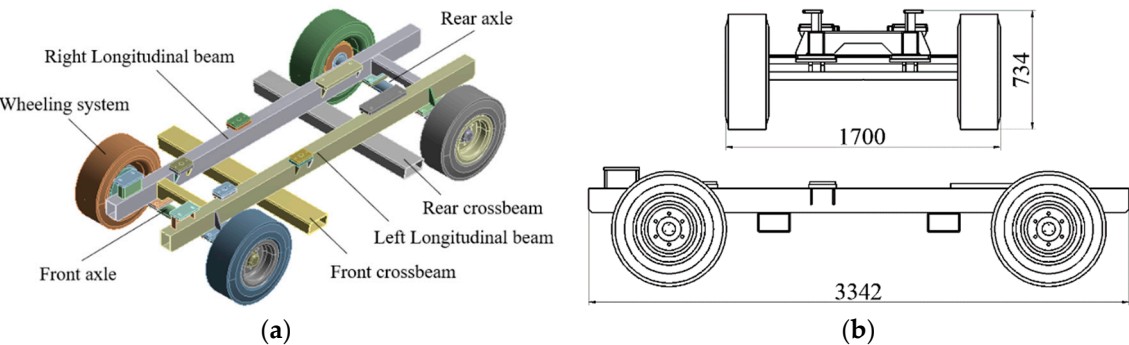

**Figure 1.** Car body model. (**a**) Three-dimensional solid model of vehicle body and (**b**) main dimensions of the vehicle body.

## 2.2. Calculation Method

The finite element simulation calculation of this article was conducted using Ansys Workbench. According to the characteristics of the car body model, a combination of quadrilateral and triangular elements was used for mesh division, and some components were encrypted with mesh. Before the simulation started, this article underwent a lot of grid division and calculation. When the maximum displacement change of the vehicle frame was less than 2%, it was considered that the calculation was correct. Finally, the vehicle body was divided into 612,314 units and 1,386,074 nodes, and the meshing of the vehicle body is shown in Figure 2.

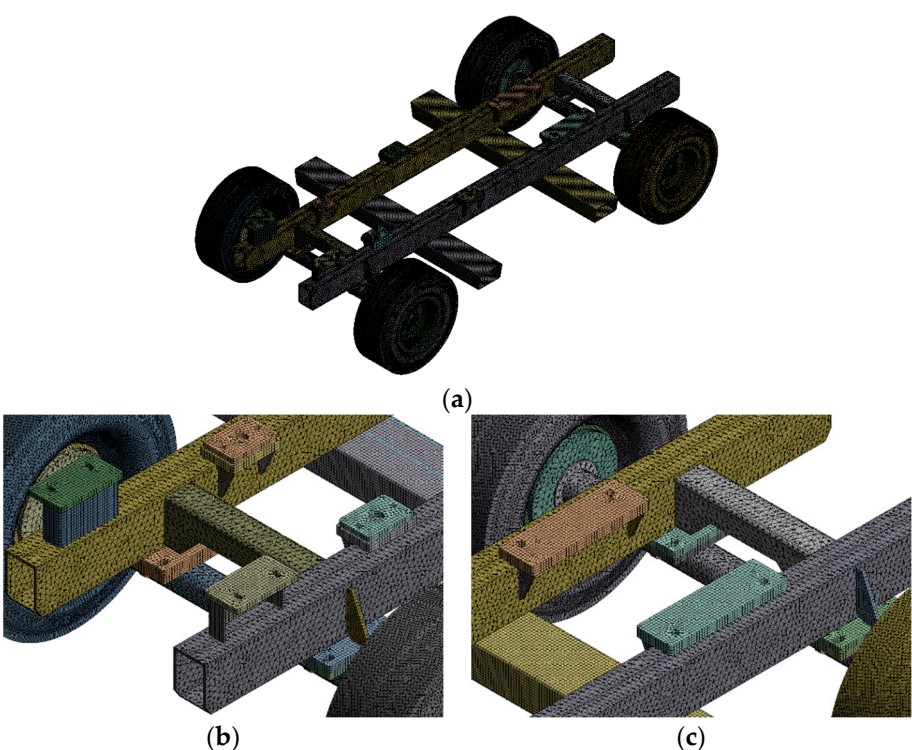

**Figure 2.** Meshing of vehicle body. (**a**) Overall meshing, (**b**) local meshing of the front axle, and (**c**) local meshing of the rear axle.

Based on the relevant literature and production processes [18–20], the material of the beam of the frame was Q345 low alloy structural steel. The material of the axle, axle bracket, and support plate was 40Cr, the material of the tire was car rubber, and the other parts were Q234 carbon structural steel; the material parameters are shown in Table 1, and the grid number of each part is shown in Table 2. Various steel grades were used to manufacture the

axle, brackets, plate, and supporting beams to meet the demand of the structure integrity of the vehicle.

**Table 1.** Frame material parameters.

| Name | Material | Density/(g·cm$^{-3}$) | Poisson's Ratio | Elastic Modulus/GPa | Yield Strength/MPa |
|---|---|---|---|---|---|
| Beam | Q345 | 7.85 | 0.2 | 206 | 345 |
| Axles, axle supports, support plates | 40Cr | 7.85 | 0.3 | 211 | 785 |
| Tires | Rubber | 1.2 | 0.47 | $7.8 \times 10^{-3}$ | - |
| Other parts | Q235 | 7.85 | 0.3 | 210 | 235 |

**Table 2.** Number of grids in each section.

| Name | Number of Grids | Name | Number of Grids |
|---|---|---|---|
| Longitudinal beam | 84,452 | Axle | 59,010 |
| Crossbeam | 27,380 | Axle support components | 21,426 |
| Wheel system | 394,318 | Other parts | 25,728 |

### 2.3. Calculated Solutions

As the main load-bearing system of the whole pump truck, the frame will bear a lot of load during the driving process. For example, the frame is subject to gravitational loads from on-board equipment such as diesel engines and self-priming pumps. During emergency turning and emergency braking, the frame will also be subjected to load forces caused by changes in acceleration. The on-board equipment of the mobile pump truck mainly includes a diesel engine and self-priming pump. The weight of the diesel engine system was 600 kg, and the weight of the self-priming pump system was 500 kg.

### 2.4. Boundary Conditions

This paper mainly carried out simulation experiments on the four working conditions of the frame: full load bending, full load torsion, emergency turning, and emergency braking. The schematic diagram of each working condition is shown in Figure 3. In the simulation process, it was necessary to define the contact types of various components of the frame. The contact types were divided into bonded, no separation, frictionless, rough, and friction types. Due to the welding fixation of various parts of the frame, the bonded type was used in the simulation process, and the difference in contact types is shown in Table 3.

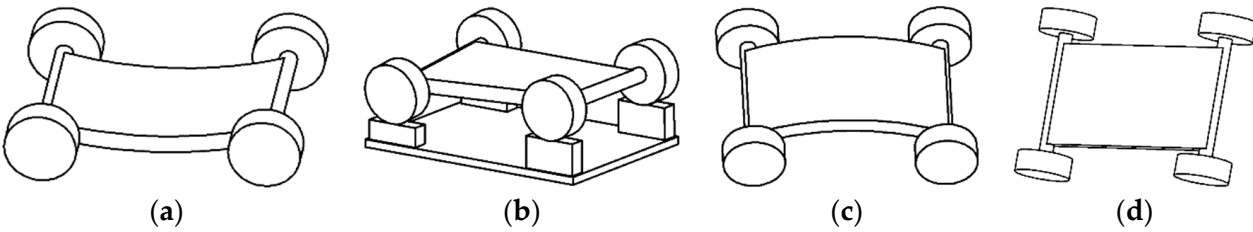

(**a**)     (**b**)     (**c**)     (**d**)

**Figure 3.** Schematic diagram of each working condition. (**a**) Bending conditions, (**b**) torsional conditions, (**c**) turning conditions, and (**d**) braking conditions.

**Table 3.** Differences in contact types.

| Contact Type | Normal Separation | Tangential Separation |
|---|---|---|
| Bonded | No | No |
| No separation | No | Yes |
| Frictionless | Yes | No |
| Rough | Yes | Yes |
| Friction | Yes | Yes |

The full load bending condition is the operating condition when the vehicle is driven on a flat road. Under this condition, the uniform and stable contact between the wheels and the road is maintained. At this time, the force of the car body in the vertical direction is mainly considered, and the influence of other factors such as lateral wind and inertial force is ignored. When setting the boundary conditions, the translational degrees of freedom in the UX, UY, and UZ directions of the four wheels before and after the car body are constrained, and the rotational degrees of freedom in the ROTX, ROTY, and ROTZ directions are constrained (transverse: X, vertical: Y, longitudinal: Z; U is the translational degree of freedom, ROT is the rotational degree of freedom, and the rest of the working conditions are the same).

The full load torsion condition is the condition in which a wheel of a vehicle is suspended under severe road conditions. The unevenness of the road surface causes asymmetric support of the frame, which results in torsion. Under the condition of full load torsion, the wheel cannot maintain uniform and stable contact with the road surface, and so the torsion condition mainly simulates a situation whereby the left front wheel suddenly hangs in the air during the driving process. When setting the boundary conditions, all of the degrees of freedom of the left front wheel are released and all of the degrees of freedom of the right front wheel are constrained. At the same time, the vertical degrees of freedom in the UY direction of the two rear wheels are constrained and all of the other degrees of freedom are released.

The emergency turning condition means that when the vehicle changes its driving direction, each part of the frame will cause displacement and stress changes due to centripetal acceleration. In order to simulate the state change in the frame under turning conditions, a gravitational acceleration of 1 g and a left-handed acceleration of 0.4 g were applied to the frame in this paper [21]. When setting the boundary conditions, the three translational degrees of freedom of the front and rear four wheels UX, UY, and UZ were constrained. At the same time, the rotational degrees of freedom of the four wheels in the ROTX, ROTY, and ROTZ directions were released.

Under emergency braking conditions, the frame will be subjected to a longitudinal inertia force load, resulting in displacement and stress changes. In order to simulate the state change in the frame under braking conditions, a gravitational acceleration of 1g and a deceleration acceleration of 0.45 g were applied to the frame in this paper. When setting the boundary conditions, the three directions of translational degrees of freedom UX, UY, and UZ of the two front-end wheels were constrained, and the rotational degrees of freedom of the ROTX, ROTY, and ROTZ directions were released. The vertical degrees of freedom UY and longitudinal degrees of freedom UZ of the two wheels at the rear end were constrained, and the other degrees of freedom were released. Table 4 shows the setting of boundary conditions for each working condition.

The front and rear axles, front and rear crossbeams, and left and right longitudinal beams of the frame were all hollow structures. In this paper, monitoring paths (1: start point, 2: end point) were added at the midline of the upper surface of these six main components. The schematic diagram of the monitoring paths is shown in Figure 4.

**Table 4.** Setting of boundary conditions for each working condition.

| Work Condition | Left Front Wheel | Right Front Wheel | Left Rear Wheel | Right Rear Wheel |
|---|---|---|---|---|
| Full load bending | All | All | All | All |
| Full load reversal | - | All | UY | UY |
| Emergency turns | UXUYUZ | UXUYUZ | UXUYUZ | UXUYUZ |
| Emergency braking | UXUYUZ | UXUYUZ | UYUZ | UYUZ |

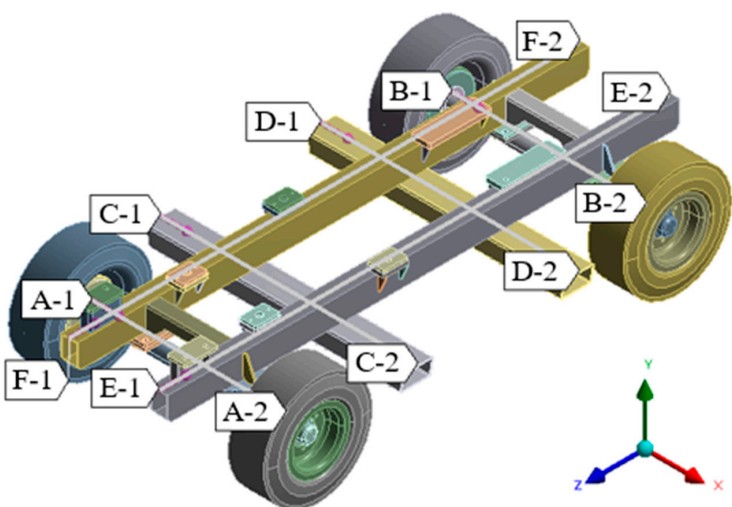

**Figure 4.** Schematic diagram of the monitoring path.

## 3. Analysis of Results

### 3.1. Full Load Bending Working Condition

The displacement calculation results of the full load bending condition of the frame are shown in Figure 5. It can be seen that the displacement in the frame gradually becomes larger from the two ends of the frame to the middle, the maximum displacement occurs in the middle of the left and right longitudinal beams ($l$ = 1671 mm), and the maximum displacement is 2.11 mm. The main reason for this deformation is that the longitudinal beam is subjected to the gravity load of the diesel engine and the self-priming pump and is mainly concentrated in the middle section; so, the middle of the longitudinal beam is deformed by downward bending.

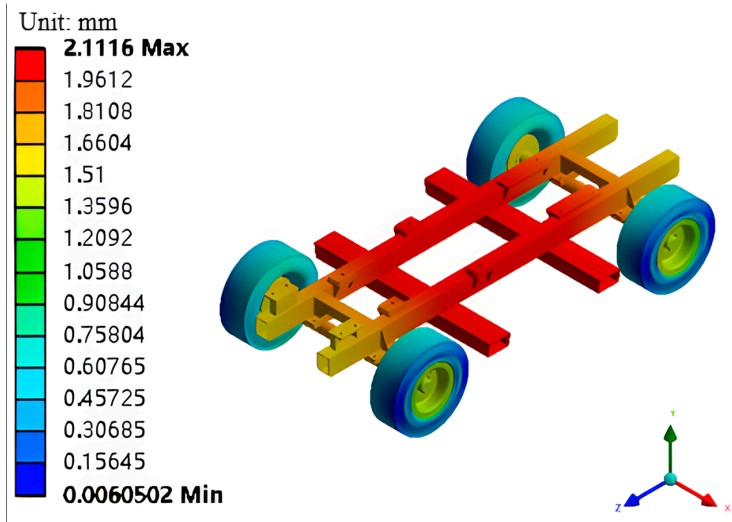

**Figure 5.** Displacement nephogram under bending condition.

As can be seen from Figure 6a, the right end of the front and rear axles (namely A-1 and B-1) is taken as the starting point, and the displacement in the front axles gradually rises from 1.57 mm to the maximum displacement of 1.82 mm in the middle, and then decreases with the increase in the distance. The displacement trend of the rear axle is consistent with that of the front axle, but the overall displacement in the rear axle is slightly larger than that of the front axle. Compared with the front axle, the minimum displacement in the rear axle is 1.56 mm, and the maximum displacement is 1.84 mm.

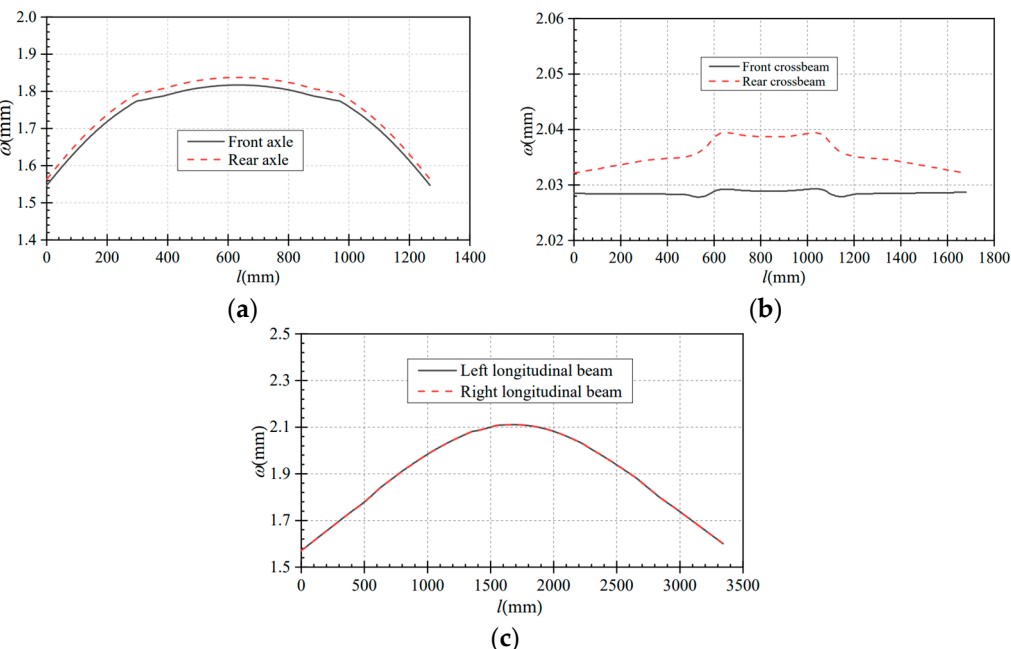

**Figure 6.** Displacement curves of main components under bending conditions. (**a**) Axle displacement curve, (**b**) crossbeam displacement curve, and (**c**) longitudinal beam displacement curve.

As can be seen from Figure 6b, the maximum displacement positions of both the front and rear crossbeams occur at the contact between the crossbeams and the inner side of the left and right longitudinal beams (namely $l = 630$ mm and $l = 1050$ mm). The displacement on both sides of the front crossbeam was 2.0285 mm. In the area of contact between the front crossbeam and the outer side of the left and right longitudinal beams, the displacement in the front crossbeam decreases and rises slightly, where the minimum displacement is 2.0278 mm and the maximum displacement is 2.0292 mm. As the rear crossbeam is close to the installation position of the self-priming pump, the overall displacement in the rear crossbeam is slightly larger than the front crossbeam. The minimum displacement in the rear crossbeam is located on both sides of it, and the minimum displacement is 2.0322 mm. The displacement in the rear crossbeam continues to increase from both sides to the middle to the maximum displacement, and the maximum displacement is 2.0394 mm.

As can be seen from Figure 6c, taking the front end of the longitudinal beams (namely E-1 and F-1) as the starting point, the displacement variation trend of the left and right longitudinal beams is consistent. The displacement changes from the front end of the longitudinal beam to the rear end of the longitudinal beam, showing the phenomenon of increasing first and then decreasing, and the maximum displacement occurs at the middle position of the longitudinal beam ($l = 1671$ mm). Among them, the displacement in the front end of the left and right longitudinal beams is 1.57 mm, the maximum displacement is 2.11 mm, and the displacement in the end of the longitudinal beam is 1.60 mm. The overall displacement in the frame and the displacement in each key component are small in the bending condition.

As can be seen from Figure 7, the maximum stress location of the frame occurs on the axle bracket at the connection between the base of the left and right longitudinal beams

and the axle, and the maximum stress is 71.76 MPa. According to the yield strength of the frame material, the safety factor of each part of the frame is 10.94. It can be seen that the strength of the frame meets the design requirements, and there is a certain design margin.

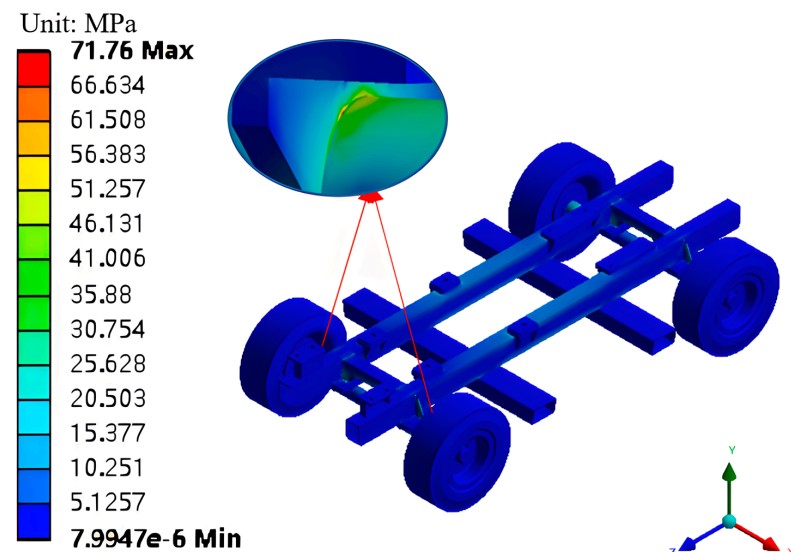

**Figure 7.** Stress nephogram under bending condition.

As can be seen from Figure 8a, the stresses in the front and rear axles change from the ends of the axles to the middle with a trend of increasing first and then decreasing, and they increase rapidly when approaching the outer axle bracket. The maximum stresses at the midline of the upper surface of the front and rear axles are at $l = 297$ mm and $l = 971$ mm, with maximum stresses of 51.50 MPa and 52.80 MPa, respectively. During the stress increase from both ends of the axle to the outer axle bracket, a small amplitude of stress fluctuation occurs at $l = 225$ mm and $l = 1043$ mm. At the same time, under the action of the inner and outer axle brackets, there are two stress fluctuations in the process of stress drop from the outer axle bracket to the middle of the axle.

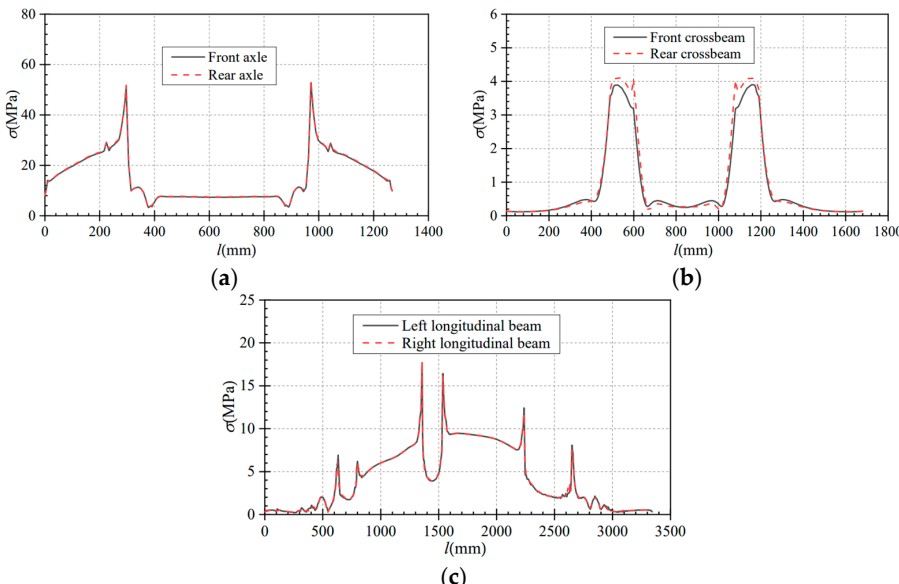

**Figure 8.** Stress curves of main components under bending condition. (**a**) Axle stress curve, (**b**) crossbeam stress curve, and (**c**) longitudinal beam stress curve.

As can be seen from Figure 8b, the maximum stress position of the front and rear crossbeams occurs at the connection location between the longitudinal beams and the crossbeams. The maximum stress of the front crossbeam is 3.90 MPa, and the maximum stress of the rear crossbeam is 4.11 MPa. The maximum stress in the rear crossbeam is slightly larger than that of the front crossbeam, while the stress in the rear crossbeam increases suddenly at $l$ = 599 mm and $l$ = 1080 mm, mainly because the rear crossbeam is close to the installation position of the self-priming pump, and the concentrated load received is also relatively large, resulting in stress fluctuations at the inner contact between the rear crossbeam and the left and right longitudinal beam connections.

Figure 8c represents the stress trend of the midline on the upper surface of the left and right longitudinal beams. From the front end of the longitudinal beam to the rear end of the longitudinal beam, the stress changes show an overall trend of first increasing and then decreasing. In the three intervals of $l$ = 633 mm to $l$ = 799 mm, $l$ = 1357 mm to $l$ = 1536 mm, and $l$ = 2237 mm to $l$ = 2650 mm, the stress of the longitudinal beam appears to decrease significantly and then rebound. This is mainly because the diesel engine and self-priming pump are fixed to the longitudinal beam via the connector, which reduces the stress at the contact area between the longitudinal beam and the bottom of the connector. The stress value at the contact area between the front and rear ends of the connector and the longitudinal beam is larger, which is consistent with the actual situation under bending conditions.

### 3.2. Full Load Torsional Working Condition

As can be seen from Figure 9, when the left front wheel is suspended, the left front area of the frame will be subjected to more gravity loads due to the constraint asymmetry. The maximum displacement in the frame occurs at the left front wheel position, and the trend of displacement changes gradually: it decreases from the left front wheel to the right rear wheel, and the maximum displacement is 18.18 mm.

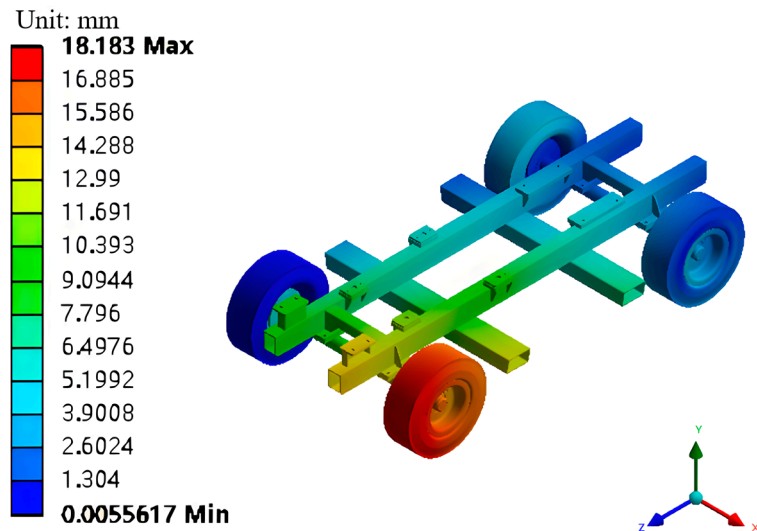

**Figure 9.** Displacement nephogram under torsional conditions.

As can be seen from Figure 10a, the displacement in the front axle continuously increases from 4.87 mm at the right end of the axle (namely A-1) to 14.85 mm at the left end because the left front wheel is suspended. Compared with the front axle, the displacement change trend of the rear axle fluctuates less, and the displacement change in the rear axle increases from 1.32 mm at the right end of the axle to 2.94 mm at the maximum displacement, and then decreases to 2.93 mm at the left end of the axle.

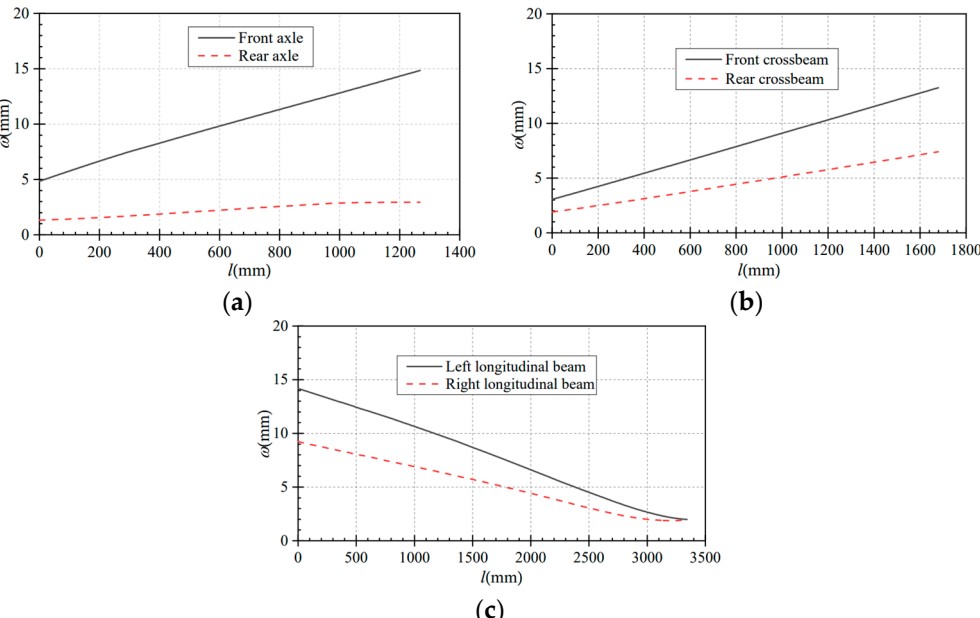

**Figure 10.** Displacement curves of main components under torsion condition. (**a**) Axle displacement curve, (**b**) crossbeam displacement curve, and (**c**) longitudinal beam displacement curve.

As can be seen from Figure 10b, the displacement changes in both the front and rear crossbeams show an increasing trend, with the right end of the crossbeam (namely C-1 and D-1) as the starting point. The displacement variation in the front crossbeam is larger than that in the rear crossbeam. The displacement change in the front crossrail increases from 3.05 mm to 13.25 mm, and the displacement change in the rear crossrail increases from 1.91 mm to 7.40 mm.

As can be seen from Figure 10c, taking the front end of the longeron (namely E-1 and F-1) as the starting point, the displacement variation trend of the left and right longitudinal beam is consistent. The displacement change from the front of the longitudinal beam to the end of the longitudinal beam decreases first and then increases, and the displacement change in the left longitudinal beam is larger than that of the right beam; the displacement change in the left longitudinal beam is larger than that in the right crossbeam. This phenomenon is realized because the support state of the left front wheel changes, making the end of the longitudinal beam produce a slight displacement in the rear axle as the pivot point. The gravity of the vehicle equipment makes the displacement change in the frame when the left front wheel is suspended, and the high elasticity of the rubber material magnifies the displacement change to a certain extent.

Combined with Figures 9 and 10, the maximum displacement in the left half of the frame under the torsion condition is 18.18 mm, while the displacement in the right half of the frame is less than 10 mm, and the displacement in the left side of the frame is larger than the displacement in the right side. It can be seen that the torsion condition is a typical dangerous condition in the process of vehicle driving, and so it should be avoided as much as possible in the process of driving.

As can be seen from Figure 11, due to the asymmetry of the frame support caused by the overhang of the left front wheel, the maximum stress of the frame occurs at the front support plate of the left longitudinal beam, and the maximum stress is 352.68 MPa. The yield strength of the support plate is 785 MPa. It can be calculated that the safety factor of the frame under the full load torsion condition is 2.23. The strength of the frame meets the design requirements, and there is a certain design margin.

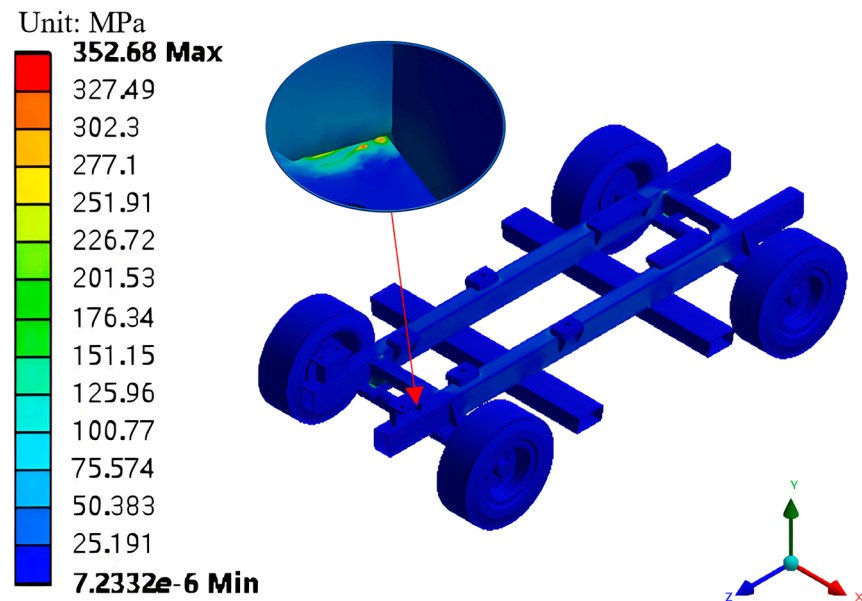

**Figure 11.** Stress nephogram under torsional conditions.

As can be seen from Figure 12a, the stresses in the front and rear axles have opposite trends. Taking the right end of the axle (namely A-1 and B-1) as the starting point, the stresses in the front axle showed a general trend of increasing first and then decreasing. The stress increases rapidly when it is close to the outer axle bracket, which means that the maximum stress position at the midline of the upper surface of the front axle is at $l = 297$ mm, and the maximum stress is 107.04 MPa. When the stress curve crosses the highest point, the stress decreases rapidly from 107.04 MPa at $l = 297$ mm to 25.11 MPa at $l = 315$ mm, and then decreases slowly. Due to the action of the inner and outer axle support, the stress curve will show a small range of stress fluctuations in the process of falling.

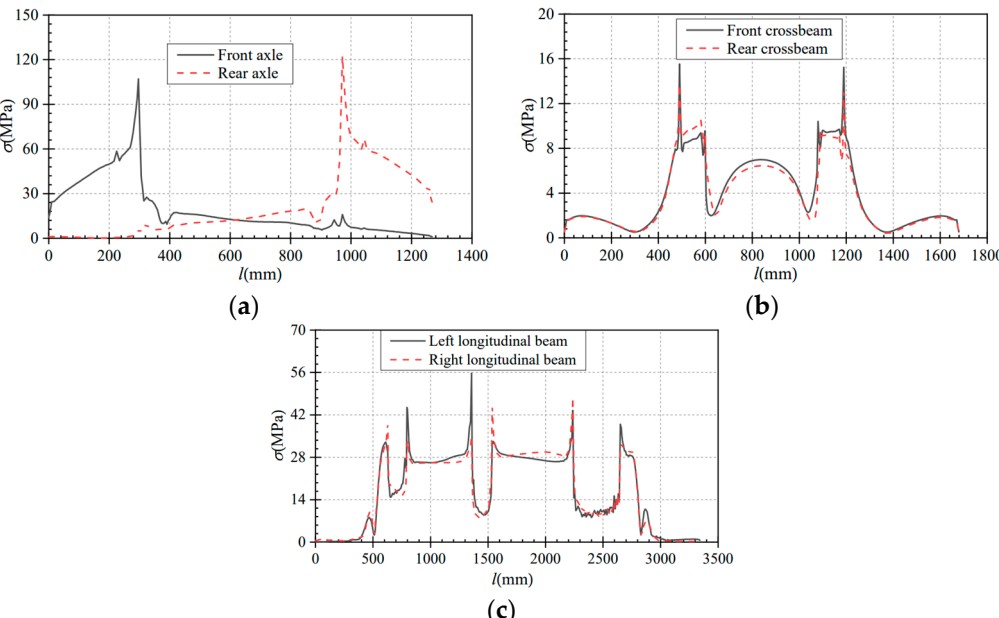

**Figure 12.** Stress curves of main components under torsion condition. (**a**) Axle stress curve, (**b**) crossbeam stress curve, and (**c**) longitudinal beam stress curve.

As can be seen from Figure 12b, the maximum stress position of the front and rear crossbeams occurs at the contact position between the crossbeam and the outer side of the longitudinal beam (at *l* = 490 mm and *l* = 1189 mm), and then decreases rapidly. The maximum stress of the front crossbeam and rear crossbeam is 15.53 MPa and 13.71 MPa, respectively. The stress fluctuates in a certain range at the interface between the crossbeam and the longitudinal beam. In the middle of the crossbeam, the maximum stress of the front crossbeam is larger than that of the rear front crossbeam.

Figure 12c represents the stress variation trend in the midline of the upper surface of the left and right longitudinal beams, and the stress variation fluctuation is obvious. At the connection between the longitudinal beam and the other components, the stress appears to be obviously reduced and then rebounded, and the change trend of the left and right longitudinal beams is approximately equal. Due to the asymmetric support of the frame, there are differences in the individual stress peaks between the left and right longitudinal beams, and the maximum stress value of the left longitudinal beam is higher than that of the right longitudinal beam, with the maximum stresses of the left longitudinal beam and right longitudinal beam being 55.74 MPa and 47.77 MPa, respectively. The maximum stress at each part of the frame is much less than the allowable stress of its material, and so the strength of the frame under torsional conditions meets the design requirements.

### 3.3. Emergency Turning Conditions

As can be seen from Figure 13, the maximum displacement in the turning condition occurs at the wheel, and the maximum displacement is 2.61 mm. When the wheels are not considered, the displacement change in the frame becomes larger from both ends to the middle in the state of turning left. The maximum displacement occurs at the middle of the longitudinal beam on the right side of the frame, and the maximum displacement in the middle section of the frame is 2.20 mm. At the same time, the maximum displacement area of the middle section of the frame tends to increase from the left side to the right side, which is consistent with the actual situation of turning conditions.

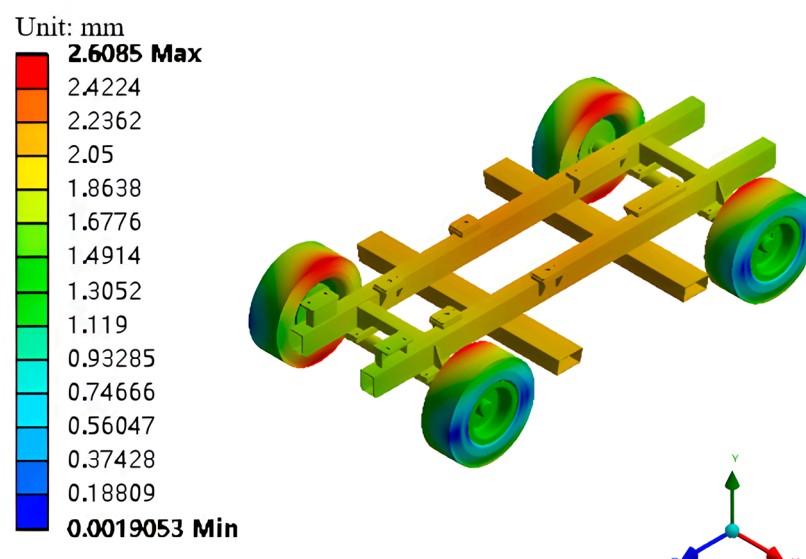

**Figure 13.** Displacement nephogram under turning condition.

Figure 14a shows that the displacement in the front axle rises gradually from 1.61 mm at the right end of the axle (A-1) to the maximum displacement of 1.89 mm in the middle of the axle; after that, the displacement gradually decreases as the distance increases, and the displacement in the right end of the axle is slightly larger than the displacement in the left end, which is 1.59 mm. The displacement trend of the rear axle is consistent with that of the front axle. Compared with the front axle, the displacement in the right end of the rear

axle was 1.62 mm, the maximum displacement was 1.91 mm, and the displacement in the left end was 1.60 mm.

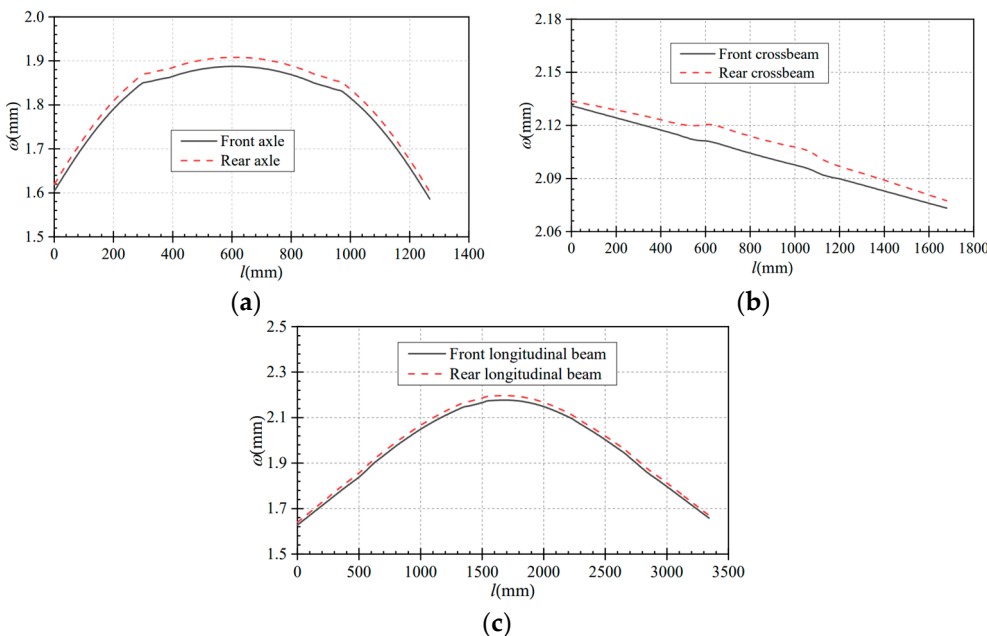

**Figure 14.** Displacement curves of main components under turning condition. (**a**) Axle displacement curve, (**b**) crossbeam displacement curve, and (**c**) longitudinal beam displacement curve.

As can be seen from Figure 14b, the displacement curves of the front and rear crossbeams show a continuous decrease when the right end of the crossbeams (namely C-1 and D-1) is taken as the starting point. The displacement in the front and rear crossbeams decreased from 2.13 mm and 2.13 mm, respectively, to 2.07 mm and 2.08 mm. The displacement in the right side of the front and rear crossbeams is larger than that of the left side, which is mainly caused by the centripetal acceleration during the turning process.

As can be seen from Figure 14c, taking the front end of the longeron (namely E-1 and F-1) as the starting point, the displacement variation trend of the left and right longerons is consistent. The displacement change from the front end of the longitudinal beam to the rear end of the longitudinal beam shows the phenomenon of first increasing and then decreasing, and the overall displacement in the right longitudinal beam is slightly larger than the overall displacement in the left longitudinal beam. Among them, the front end displacements in the left and right longitudinal beams are 1.63 mm and 1.64 mm, and the maximum displacements occur in the middle of the longitudinal beams. The maximum displacement in the left and right longitudinal beams is 2.18 mm and 2.20 mm, and the displacements in the right end are 1.66 mm and 1.67 mm, respectively. The overall displacement in the frame and the displacement in each key component are small in the turning condition.

As can be seen from Figure 15, the maximum stress location of the frame under the turning condition occurs on the axle bracket at the connection between the base of the right longitudinal beam and the front axle, and the maximum stress is 79.718 MPa. The yield strength of the axle support is 785 MPa. It can be calculated that the safety factor of the frame under the emergency turning condition is 9.85, and so the strength of the frame meets the design requirements.

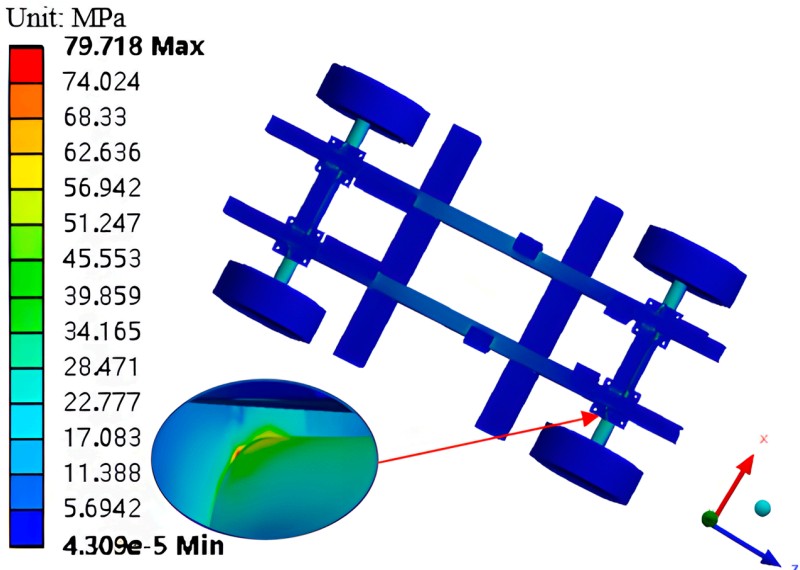

**Figure 15.** Stress nephogram under turning condition.

As can be seen from Figure 16a, the stresses on the front and rear axles change from the ends of the axles to the middle with the trend of first increasing and then decreasing, and the stresses increase rapidly when approaching the axle supports, which means that the maximum stresses on the midline of the upper surface of the front and rear axles are located at $l$ = 297 mm and $l$ = 971 mm. The maximum stress on the front axle is 57.62 MPa, and the maximum stress on the rear axle is 58.25 MPa. In addition, the stress at $l$ = 297 mm is slightly greater than the stress at $l$ = 971 mm, which is mainly caused by the centrifugal force generated in the process of turning left. During the stress increase from both ends of the axle to the outer axle bracket, a small amplitude of stress fluctuation occurs at $l$ = 225 mm and $l$ = 1043 mm. At the same time, due to the inner and outer axle brackets, there are two stress fluctuations in the process of stress dropping from the outer axle support to the middle position of the axle.

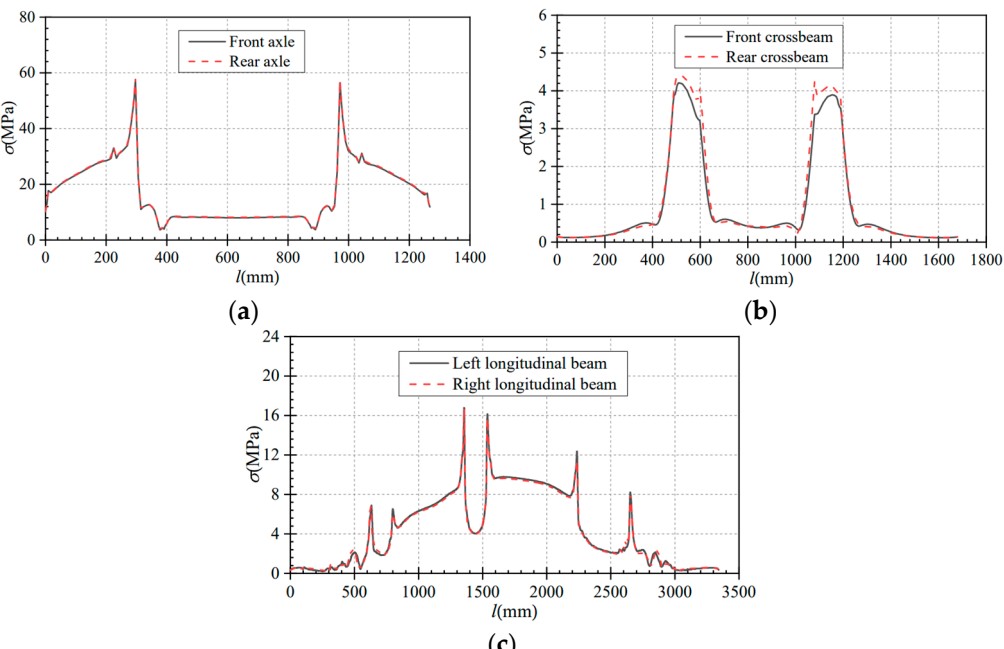

**Figure 16.** Stress curves of main components under turning condition. (**a**) Axle displacement curve, (**b**) crossbeam displacement curve, and (**c**) longitudinal beam displacement curve.

As can be seen from Figure 16b, the maximum stress position of the front and rear crossbeams occurs at the contact position between the longitudinal beam and the crossbeam, and the stress peaks on the right side of the front and rear crossbeams are higher than those of the left side, which is also caused by the centripetal acceleration during the turning process. The maximum stress of the front crossbeam is 4.21 MPa, and that of the rear crossbeam is 4.39 MPa. In addition, the stress in the rear crossbeams increases suddenly at $l = 599$ mm and $l = 1080$ mm, mainly because this is due to the fact that the rear crossbeam is close to the self-priming pump mounting position, and the concentrated load is relatively large. A small range of stress fluctuation is generated at the inner contact of the rear crossbeam connected to the left and right longitudinal beams.

From Figure 16c, it can be seen that from the front end of the longitudinal beam to the rear end of the longitudinal beam, the stress variation in the longitudinal beam shows an overall trend of increasing first and then decreasing. In the three intervals from $l = 633$ mm to $l = 799$ mm, from $l = 1357$ mm to $l = 1536$ mm, and from $l = 2237$ mm to $l = 2650$ mm, the longitudinal beam stress significantly reduces after the rising phenomenon. This is mainly because the diesel engine and self-priming pump are fixed on the longitudinal beam through the connecting parts, so that the stress at the contact surface between the longitudinal beam and the connecting piece is reduced.

### 3.4. Emergency Braking Conditions

As can be seen from Figure 17, since the wheel is in full contact with the road surface under the braking condition, the wheel is where the maximum displacement occurs, and the maximum displacement is 2.81 mm. In addition, without considering the wheels, the displacement deformation of the frame gradually increases from the front and rear ends of the frame to the middle. The reason for this deformation is that the gravity load of the diesel engine and the self-priming pump is mainly concentrated in the middle of the frame, where the inertia force load is the largest, and the displacement deformation is also large.

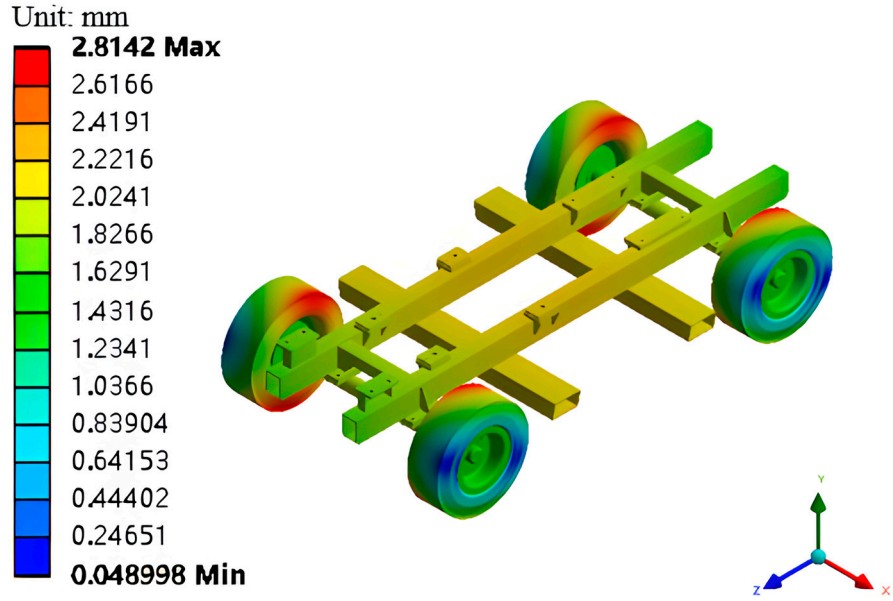

**Figure 17.** Displacement nephogram under braking condition.

From Figure 18a, taking the right end of the axle (A-1 and B-1) as the starting point, the displacement in the front axle gradually rises from 1.63 mm to reach the maximum displacement of 1.94 mm in the middle, and after which the displacement gradually decreases with the increasing distance. The displacement change trend of the rear axle is consistent with the front axle. Compared with the front axle, the displacement in the right end of the axle of the rear axle is 1.59 mm and the maximum displacement is 1.89 mm.

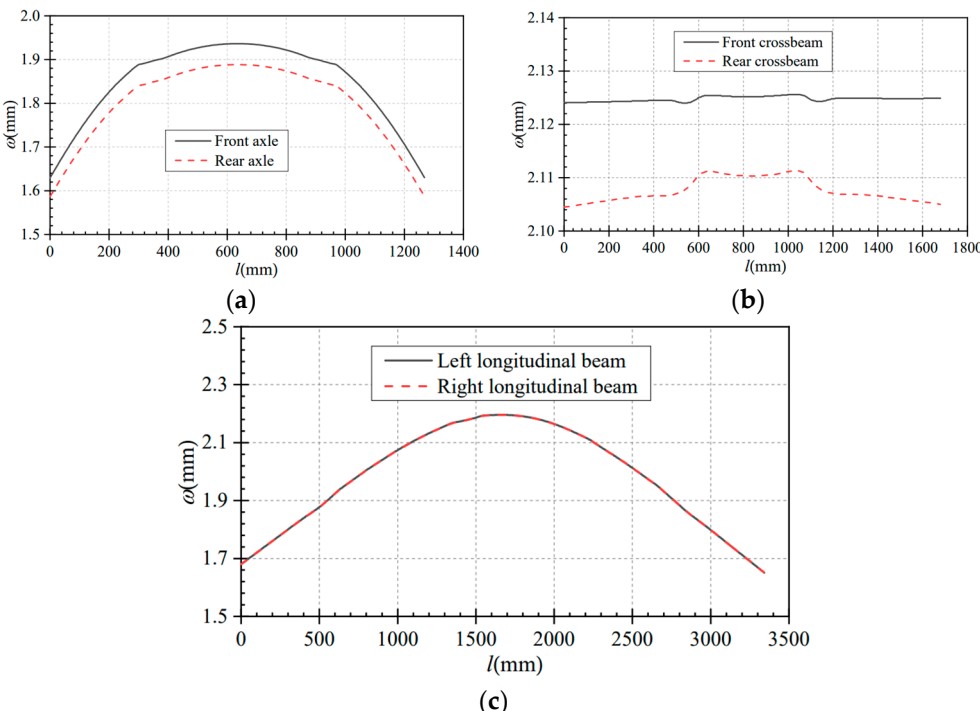

**Figure 18.** Displacement curves of main components under braking condition. (**a**) Axle displacement curve, (**b**) crossbeam displacement curve, and (**c**) longitudinal beam displacement curve.

From Figure 18b, it can be seen that the maximum displacement position of both the front and rear crossbeams occurs at the contact between the crossbeams and the inner side of the left and right longitudinal beams (namely $l = 630$ mm and $l = 1050$ mm). At the same time, the overall displacement in the front crossbeam is slightly larger than that of the rear crossbeam due to the inertia force during the braking process. The maximum displacement in the front crossbeam is 2.126 mm and the maximum displacement in the rear crossbeam is 2.111 mm. The displacement on both sides of the front crossbeam is 2.124 mm. In the contact area between the front crossbeam and the outer side of the left and right longitudinal beam, the displacement in the front crossbeam decreases and rises slightly. The minimum displacement in the rear crossbeam is located on both sides of it, and the minimum displacement is 2.105 mm. The displacement in the rear beam continuously increases from both sides to the middle to the maximum displacement.

As can be seen from Figure 18c, taking the front end of the longitudinal beams (namely E-1 and F-1) as the starting point, the displacement variation trend of the left and right longitudinal beam is consistent, and the displacement variation from the front end of the longeron to the back end of the longeron increases first and then decreases. The front end displacement in the left and right longitudinal beams is 1.68 mm and 1.68 mm, respectively, and the maximum displacement occurs in the middle of the longitudinal beam; the maximum displacements in the middle of the left and right longitudinal beams are 2.20 mm and 2.20 mm, and the displacements in the rear end are 1.65 mm and 1.65 mm, respectively. The overall displacement in the frame and the displacement in each key component are small under the braking condition.

As can be seen from Figure 19, the maximum stress location of the frame under braking conditions is the axle bracket at the connection between the base of the left and right longitudinal beams and the axle, and the maximum stress is 74.907 MPa. Combined with the yield strength of the longitudinal beam support frame material, it can be calculated that the safety factor of the frame under the emergency braking condition is 10.48, which can meet the strength design requirements and has a certain margin.

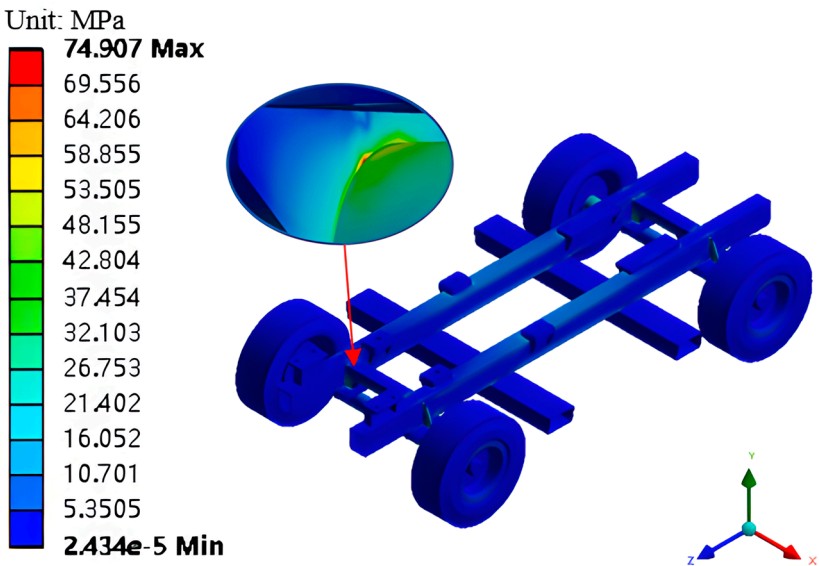

**Figure 19.** Stress cloud diagram under braking condition.

From Figure 20a, the stress of the front and rear axles increases first and then decreases from both ends of the axle to the middle, and increases rapidly when approaching the axle support, which means that the maximum stresses on the midline of the upper surface of the front and rear axles are located at $l$ = 297 mm and $l$ = 971 mm, and the maximum stresses are 57.25 MPa and 58.64 MPa, respectively.

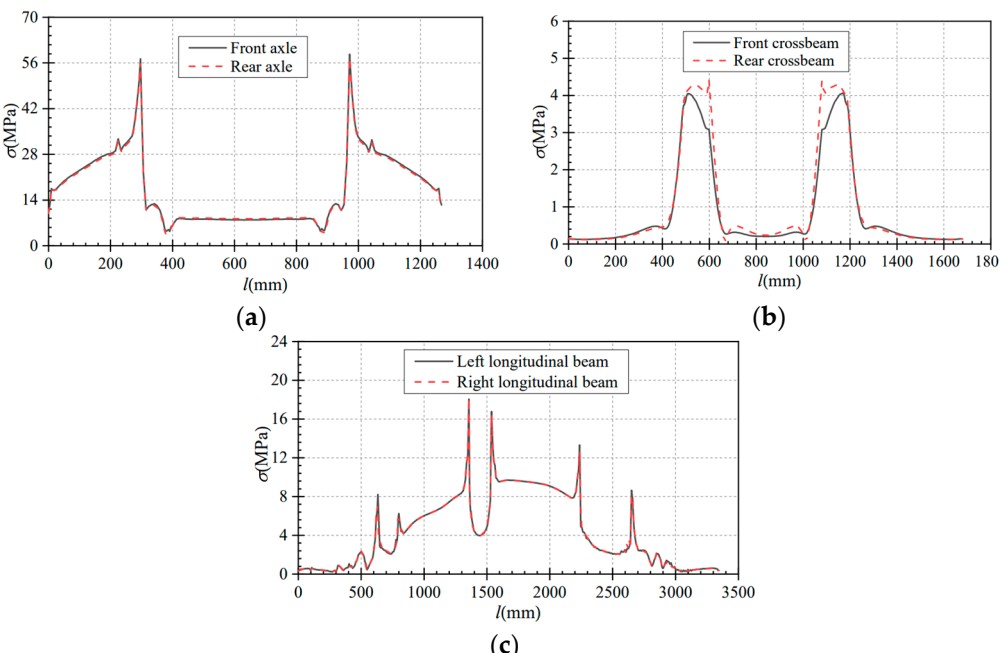

**Figure 20.** Stress curves of main components under braking condition. (**a**) Axle displacement curve, (**b**) crossbeam displacement curve, and (**c**) longitudinal beam displacement curve.

As can be seen from Figure 20b, the maximum stress position of the front and rear crossbeams occurs at the connection position between the longitudinal beams and the crossbeams, and the maximum stress of the rear crossbeam is slightly larger than the maximum stress of the front crossbeam; the maximum stress of the front crossbeam is 4.06 MPa and the maximum stress of the rear crossbeam is 4.42 MPa.

From Figure 20c, it can be seen that the stress change trend under the braking condition is similar to the stress change trend under the bending condition. From the front end of the longitudinal beam to the rear end of the longitudinal beam, the overall trend of stress change in the longitudinal beam first increases and then decreases. The maximum stress of the left longitudinal beam is 18.06 MPa and the maximum stress of the right longitudinal beam is 18.16 MPa. In the three intervals from $l = 633$ mm to $l = 799$ mm, $l = 1357$ mm to $l = 1536$ mm, and $l = 2237$ mm to $l = 2650$ mm, the stresses in the longitudinal beams appear to significantly decrease and then rebound. The stresses in all of the parts of the frame under the braking condition are less than 100 MPa, which is much less than the yield strength limit of each material, and so the strength of the frame under the braking condition meets the design requirements.

Based on the above analysis, it can be concluded that the maximum stress and maximum displacement generated by the mobile pump truck under four typical working conditions can meet the requirements of frame strength design. Among them, the maximum stress and maximum displacement under the full load bending condition are 71.76 MPa and 2.11 mm; the maximum stress and maximum displacement under the full load torsion condition are 352.68 MPa and 18.18 mm; the maximum stress and maximum displacement under emergency turning conditions are 79.718 MPa and 2.68 mm; and the maximum stress and maximum displacement under emergency braking conditions are 74.907 MPa and 2.81 mm. It can be analyzed that the safety factor of the frame under four typical working conditions is greater than 2.0, which provides a reference for the design of the subsequent mobile pump truck frame. The safety factors under four typical working conditions are shown in Table 5.

**Table 5.** Safety factor under four typical working conditions.

| Type | Full Load Bending | Full Load Torsion | Emergency Turning | Emergency Braking |
|---|---|---|---|---|
| Safety factor | 10.94 | 2.23 | 9.85 | 10.48 |

## 4. Conclusions

As the main bearing part of the mobile pump truck, it is of great significance to study the strength of the frame to improve the safety and reliability of the whole vehicle. In this paper, the static characteristics of the frame model under classical working conditions were obtained via the numerical simulation of four working conditions of the frame: full load bending, full load torsion, emergency turning, and emergency braking. The main conclusions are as follows:

(1) Through the analysis of displacement nephogram and stress nephogram under four working conditions, it can be obtained that the maximum displacement in the frame is not more than 3 mm and the maximum stress is not more than 80 MPa under the three working conditions of the frame: full load bending, emergency turning, and emergency braking. The frame can meet the strength design requirements, and there is a certain margin in the strength of the frame, which can be designed to be lightweight while ensuring the structural strength of the frame.

(2) The deformation and stress of the frame under torsion conditions are large, the maximum displacement is 18.18 mm, and the maximum stress is 352.68 MPa, but both are far below the yield strength limit of the material, and can meet the design requirements of the frame strength. However, as the most dangerous condition of the frame, the torsion condition should be avoided as far as possible in real life.

(3) Through the displacement and stress monitoring of the axle, beam, and longitudinal beam, this paper can clearly reflect the displacement and stress changes in the main parts of the body under various working conditions of the frame, and more accurately predict the structural performance of the mobile pump truck. The analysis results can provide a reference for the subsequent design of the mobile pump truck frame.

**Author Contributions:** Conceptualization, S.-P.L. and Y.-L.Z.; methodology, Y.-L.Z.; software, H.-B.L.; validation, H.-B.L. and L.C.; formal analysis, L.C.; investigation, S.-P.L.; resources, Y.-L.Z.; data curation, L.C.; writing—original draft preparation, H.-B.L.; writing—review and editing, Y.-L.Z.; visualization, L.C.; supervision, Y.-L.Z.; project administration, S.-P.L.; funding acquisition, Y.-L.Z. All authors have read and agreed to the published version of the manuscript.

**Funding:** The research was financially supported by the "Pioneer" and "Leading Goose" R&D Program of Zhejiang (grant no. 2022C03170), Science and Technology Project of Quzhou (grant no. 2022K98), Zhejiang Provincial Natural Science Foundation of China (grant no. LZY21E060001), and Science and Technology Project of Zhejiang (grant no. LGC21E050001).

**Institutional Review Board Statement:** Not applicable.

**Informed Consent Statement:** Not applicable.

**Data Availability Statement:** The data used to support the findings of this study are available from the corresponding author upon request.

**Conflicts of Interest:** The authors declare no potential conflict of interest with respect to the research, authorship, and/or publication of this article.

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
