# Peer review of "Static Analysis of Mobile Pump Truck Frame for Four Typical Working Conditions"

_applsci, doi:10.3390/app13127275_

Round 1

Reviewer 1 Report

Journal                         : Appl. Sci.

Title                               : Static analysis of mobile pump truck frame for four typical working conditions

ID                                   : applsci-2454605

Authors                       : Li et al.

Manuscript Type     : Research Original Paper

Date Reviewed        : June 2023

Dear Editor,

I would like to express my gratitude for the opportunity to review the research paper being considered for publication in Appl. Sci.. I have thoroughly examined the paper and evaluated its scientific quality and language efficiency.

The paper presents a research study utilising numerical method simulations to investigate the deformation of a mobile pump truck frame. I agree that studying these issues is crucial, and the topic aligns well with the scope of the interdisciplinary journal.

While the main text provide a clear explanation of the research background, there is a need for clearer explanations in terms of the originality, contribution to science, major aim, methodology, discussion, and conclusion sections. The analysis methodology is well-known, therefore, the simulation details should be precisely explained. Additionally, the research paper only cites approximately 14 scientific references, which should be increased. This study seems more like a case study or a BSc thesis rather than a research study, and it would be beneficial to make the model utilised in the study more specific to add originality to the paper.

Based on the content of the research, I believe implementing following revisions will enhance the overall quality and clarity of this paper.

1.       Revise the abstract to mention numerical results and comparison outputs. Also, include the names of all material standards utilised in the simulation.

2.       Indicate the volumetric dimensions of the truck model on the model given in Figure 1. Provide the model dimensions used in the simulation according to technical drawing rules.

3.       Provide detailed references for the material standards and properties.

4.       Provide more details about the creation of the FE model (mesh structure) in the paper. Include total element and node numbers. Mention the mesh verification issue in this section.

5.       Explain the type of contact definitions assigned in the simulation and discuss their effects on the simulation results in the boundary condition section.

6.       Present the simulation results, including maximum reaction force, deformation, and equivalent stress by parts. Provide Factor of Safety (FoS) values in a table format, including stress and deformation values.

7.       Provide descriptions and units for all parameters and equations throughout the text. Ensure clarity in describing equations, and carefully check the indication of SI units against a document providing SI unit guidance.

8.       Increase the number of scientific references cited in the research paper. Although the core methodology is well-known, further support for the methodical background should be provided.

9.       Improve the quality of graphs, schematics, and tables in the paper with more professional graphic design approaches. Ensure consistency in letters, characters, and units used in figures and tables compared to the main text for readability. Additionally, present simulation printouts with test order, including understandable numerical values and units, along with readable legends. (Provide more detailed simulation printouts).

In conclusion, AFTER THE SUGGESTED MAJOR REVISIONS ARE MADE, the reviewer recommends this paper FOR PUBLICATION in Appl. Sci.

Author Response

Reviewer: 1

Comment 1: Revise the abstract to mention numerical results and comparison outputs. Also, include the names of all material standards utilised in the simulation.

Response: Thanks for your comments. We have carefully revised the abstract to add numerical and comparative results in the first paragraph on page 1. At the same time, we have detailed all the material standard names used in the simulation in section “2.2 Calculation method”. Please check it.

------“The results show that the stress and deformation generated by the mobile pump truck under full load bending, emergency turning, and emergency braking conditions are relatively small, while they will generate significant stress and deformation under torsional conditions, but all meet the strength design requirements. Among them, the maximum stress and maximum displacement under full load bending condition are 71.76MPa and 2.11mm; the maximum stress and maximum displacement under full load torsion condition are 352.68MPa and 18.18mm; the maximum stress and maximum displacement under emergency turning conditions are 79.718MPa and 2.68mm; the maximum stress and maximum displacement under emergency braking conditions are 74.907MPa and 2.81mm. The analysis results can provide a reference basis for the design of the mobile pump truck frame in the future.”

“Based on relevant literature and production processes [18-20], the material of the beam of the frame is Q345 low alloy structural steel. The material of axle, axle bracket and support plate is 40Cr, the material of tire is car rubber, other parts are Q234 carbon struc-tural steel, the material parameters are shown in Table 1, and the grid number of each part is shown in Table 2. Various steel grades was used to manufacture the axle, brackets, plate and supporting beams to meet the demand of structure integrity of the vehicle.”

Comment 2: Indicate the volumetric dimensions of the truck model on the model given in Figure 1. Provide the model dimensions used in the simulation according to technical drawing rules.

Response: Thanks for your comments. We have carefully pointed out the relevant dimensions of the truck model and added the model dimensions used in the simulation from lines 83 to 88. Please check it.

------“The length of the 3D model of the frame is 3342mm, the width is 1700mm and the height is 734mm. In the simulation process, the model is established according to the 1:1 ratio. Considering the main factors, simplify the process holes on the non-stressed parts, crossbeams and longitudinal beams in the body structure, and simplify the chamfers and rounded corners, have little impact on the strength of the car body, so as to obtain the simplified frame model. The simplified model is shown in Figure 1.”

(a) 3D solid model of vehicle body                          (b) Main dimensions of the vehicle body

Figure 1. Car body model.

Comment 3: Provide detailed references for the material standards and properties.

Response: Thanks for your comments. We have carefully provided detailed material standards and performance references from lines 107 to 112. Please check it.

------“Based on relevant literature and production processes [18-20], the material of the beam of the frame is Q345 low alloy structural steel. The material of axle, axle bracket and support plate is 40Cr, the material of tire is car rubber, other parts are Q234 carbon structural steel, the material parameters are shown in Table 1, and the grid number of each part is shown in Table 2. Various steel grades was used to manufacture the axle, brackets, plate and supporting beams to meet the demand of structure integrity of the vehicle.”

Comment 4: Provide more details about the creation of the FE model (mesh structure) in the paper. Include total element and node numbers. Mention the mesh verification issue in this section.

Response: Thanks for your comments. We have carefully added more details about creating finite element models from lines 94 to 101. Please check it.

------“The finite element simulation calculation of this article was conducted in Ansys Workbench. According to the characteristics of the car body model, a combination of quadrilateral and triangular elements is used for mesh division, and some components are encrypted with mesh. Before the simulation started, this article underwent a lot of grid division and calculation. When the maximum displacement change of the vehicle frame was less than 2%, it was considered that the calculation was correct. Finally, the vehicle body was divided into 612314 units and 1386074 nodes and the meshing of vehicle body is shown in Figure 2.”

Comment 5: Explain the type of contact definitions assigned in the simulation and discuss their effects on the simulation results in the boundary condition section.

Response: Thanks for your comments. We have carefully supplemented the contact definition types specified in the simulation, and introduced the difference between the different contact types in the boundary conditions section from lines 126 to 130. Please check it.

------“In the simulation process, it is necessary to define the contact types of various components of the frame. The contact types are divided into Bonded, No Separation, Frictionless, Rough, and Frictiona types. Due to the welding fixation of various parts of the frame, the Bonded type is used in the simulation process,the difference in contact types is shown in Table 3.”

Table 3. Differences in contact types

Contact type

Normal separation

Tangential separation

Bonded

No

No

No Separation

No

Yes

Frictionless

Yes

No

Rough

Yes

Yes

Frictiona

Yes

Yes

Comment 6: Present the simulation results, including maximum reaction force, deformation, and equivalent stress by parts. Provide Factor of Safety (FoS) values in a table format, including stress and deformation values.

Response: Thanks for your comments. We have carefully supplement the safety factor of the frame under four typical operating conditions in lines 229 to 231 and 312 to 315 and 395 to 398 and 481 to 484, and give it in tabular form in lines 514 to 526. Please check it.

------“According to the yield strength of the frame material, the safety factor of each part of the frame is 10.94. It can be seen that the strength of the frame meets the design requirements, and there is a certain design margin.”

“The yield strength of the support plate is 785MPa. It can be calculated that the safety factor of the frame under the full load torsion condition is 2.23. The strength of the frame meets the design requirements, and there is a certain design margin.”

“The yield strength of the axle support is 785MPa. It can be calculated that the safety factor of the frame under the emergency turning condition is 9.85, so the strength of the frame meets the design requirements.”

“Combined with the yield strength of the longitudinal beam support frame material, it can be calculated that the safety factor of the frame under the emergency braking condition is 10.48, which can meet the strength design requirements and has certain margin.”

“Based on the above analysis, it can be concluded that the maximum stress and maximum displacement generated by the mobile pump truck under four typical working conditions can meet the requirements of frame strength design. Among them, the maximum stress and maximum displacement under full load bending condition are 71.76MPa and 2.11mm; The maximum stress and maximum displacement under full load torsion condition are 352.68MPa and 18.18mm; The maximum stress and maximum displacement un-der emergency turning conditions are 79.718MPa and 2.68mm; The maximum stress and maximum displacement under emergency braking conditions are 74.907MPa and 2.81mm. It can be analyzed that the safety factor of the frame under four typical working conditions is greater than 2.0, which provides a reference for the design of the subsequent mobile pump truck frame. The safety factors under four typical working conditions are shown in Table 5.”

Table 5 Safety factor under four typical working conditions

Type

Full load bending

Full load torsion

Emergency turning

Emergency braking

Safety factor

10.94

2.23

9.85

10.48

Comment 7: Provide descriptions and units for all parameters and equations throughout the text. Ensure clarity in describing equations, and carefully check the indication of SI units against a document providing SI unit guidance.

Response: Thanks for your comments. We have carefully checked the parameters of the full text to ensure the correctness of the unit.

Comment 8: Increase the number of scientific references cited in the research paper. Although the core methodology is well-known, further support for the methodical background should be provided.

Response: Thanks for your comments. We have carefully increased the number of scientific references cited in the paper in lines 53 to 56, 60 to 62, 66 to 71 and 107 to 108. Please check it.

------“Andrzej Banaszek et al. studied the impact of corrosion on the safety of hydraulic pipelines installed on product tankers and chemical tankers, and analyzed the impact of erosion or corrosion on the failure rate of load-bearing structures [9].”

[9] Andrzej Banaszek, Losiewicz Zbigniew, Wojciech Jurczak. Corrosion Influence on Safety of Hydraulic Pipelines Installed on Decks of Contemporary Product and Chemical Tankers[J]. Polish Maritime Research, 2018, 25(2): 71-77.

------“Andrzej Banaszek et al. used finite element method to study the stress effect of installation methods on the main lines on chemical storage tanks [12].”

[12] Andrzej Banaszek, Radovan Petrovic, Bartlomiej Zylinski. Finite element method analysis of pipe material temperature changes influence on line expansion loops in hydraulic installations on modern tankers[J]. Thermal Science, 2011, 15(1): 81-90.

------“Tomasz Urbaumski et al. conducted relevant research on the deformation of fixed plate edges due to butt joints, and analyzed the technical and structural parameters to evaluate the deformation shape [15]. Kirthana et al. used the finite element method to optimize the topology of the engine mounting bracket. By studying different material layouts and different designs, the optimal model is obtained through calculation, analysis and comparison of stress and weight [16].”

[15] Tomasz Urbański, Andrzej Banaszek, Wojciech Jurczak. Predictions of distortion of the fixed plate edge of the basis of designed experiment[J]. Polish Maritime Research, 2020, 27(1): 134-142.

[16] Kirthana S., Mohammed Khaja Nizamuddin. Finite Element Analysis and Topology Optimization of Engine Mounting Bracket[J]. Materials Today: Proceedings, 2018, 5(9): 19277-19283.

------“Based on relevant literature and production processes [18-20], the material of the beam of the frame is Q345 low alloy structural steel.”

[18] Imam Abdul Majid, Fajar Budi Laksono, Hendri Suryanto, et al. Structural Assessment of Ladder Frame Chassis using FE Analysis: A Designed Construction referring to Ford AC Cobra[J]. Procedia Structural Integrity, 2021, 33: 35-42.

[19] Shaik Sadikbasha, V. Pandurangan. High velocity impact response of sandwich structures with auxetic tetrachiral cores: Analytical model, finite element simulations and experiments[J]. Composite Structures, 2023, 317: 1-19.

[20] Ana Pavlović, Davide Sintoni, Giangiacomo Minak, et al. On the Modal Behaviour of Ultralight Composite Sandwich Automotive Panels[J]. Composite Structures, 2020, 248: 1-14.

Comment 9: Improve the quality of graphs, schematics, and tables in the paper with more professional graphic design approaches. Ensure consistency in letters, characters, and units used in figures and tables compared to the main text for readability. Additionally, present simulation printouts with test order, including understandable numerical values and units, along with readable legends. (Provide more detailed simulation printouts).

Response: Thanks for your comments. We have carefully checked the charts, tables and cloud images in the full text, and the numerical simulation results are readable and the values and units are clear. Please forgive me.

Reviewer 2 Report

The article deals with the strength analysis of the frame of a mobile pump truck type vehicle. Which affects the evaluation of its safety and reliability. Four typical situations of mobile pump truck movement statically were analyzed, determining the maximum stresses and strains of the mobile pump truck.  The results of the analysis can serve as a reference for designers of similar pumping system vehicle frames.

The paper is interesting suitable for publication after small additions.

1.In the literature review, there is a lack of work related to the finite element analysis of the strength of the frame support structure elements subject to corrosion ( including the frequently used tube-type support elements) . This has a significant impact on the strength of the structure as a function of time.

 It would be good to add a note in the Introduction about the impact on the strength of the analyzed load-bearing frames of the problem of fatigue damage and the impact of erosion or corrosion , on the failure rate of this type of load-bearing structures ( see, for example.

Banaszek A , Łosiewicz Z., Jurczak W.`(2018), Corrosion influence on safety of hydraulic pipelines installed on decks of contemporary product and chemical tankers, Polish Maritime Research No. 2 (98)/2018 Vol.25 pp.71-77, ISSN 1233-2585

In load-bearing structures, larger frames are often welded from plate elements. The deformations of such structures analyzed in the reviewed paper are interesting. 

It would be good to add in the Introduction an article on the prediction of deformations of this type of structures :

 Urbanski T., Banaszek A , Jurczak W. Predictions of distortion of the fixed plate edge of the basis of designed experiment, Polish Maritime Research no 1/2020 Vol.27, ISSN 1233-2585, pp.134-142     

 Please supplement

2 There is a lack in the work , at each of the four conditions of operation of the frame of a motor vehicle to provide a method of determining the maximum loads of the analyzed frame. (Load model of the analyzed frame structure in FEM with specific load values) This is very important for the stage of defining the finite element analysis process and possible validation of the obtained numerical simulations. Please supplement in the text of the article for better understanding by the reader of the conditions under which the analysis of the vehicle frame work was carried out and whether this is covered by the data from the measurements carried out on the actual car structure

Please supplement

 3 Table 1: For the first item Beam Material Q345 (low alloy structural steel) Poisson's ratio was entered a value of 05. It seems that it should be 0.3. Mistake ?  Please comment, clarify and correct if necessary.

 4. The inscriptions on the diagrams of Fig. 8, 10 , 12, 14, 16, 18 , 20 are not enough legible.. Please correct to make them more legible.

Author Response

Reviewer: 2

Comment 1: In the literature review, there is a lack of work related to the finite element analysis of the strength of the frame support structure elements subject to corrosion (including the frequently used tube-type support elements). This has a significant impact on the strength of the structure as a function of time. It would be good to add a note in the Introduction about the impact on the strength of the analyzed load-bearing frames of the problem of fatigue damage and the impact of erosion or corrosion , on the failure rate of this type of load-bearing structures.In load-bearing structures, larger frames are often welded from plate elements. The deformations of such structures analyzed in the reviewed paper are interesting. It would be good to add in the Introduction an article on the prediction of deformations of this type of structures.

Response: Thanks for your comments. We have carefully added relevant research on the impact of fatigue damage on the strength of load-bearing frames and the pre deformation of load-bearing structures in the introduction, from lines 53 to 56, 60 to 62 and 66 to 71. Please check it.

------“Andrzej Banaszek et al. studied the impact of corrosion on the safety of hydraulic pipelines installed on product tankers and chemical tankers, and analyzed the impact of erosion or corrosion on the failure rate of load-bearing structures [9].”

[9] Andrzej Banaszek, Losiewicz Zbigniew, Wojciech Jurczak. Corrosion Influence on Safety of Hydraulic Pipelines Installed on Decks of Contemporary Product and Chemical Tankers[J]. Polish Maritime Research, 2018, 25(2): 71-77.

------“Andrzej Banaszek et al. used finite element method to study the stress effect of installation methods on the main lines on chemical storage tanks [12].”

[12] Andrzej Banaszek, Radovan Petrovic, Bartlomiej Zylinski. Finite element method analysis of pipe material temperature changes influence on line expansion loops in hydraulic installations on modern tankers[J]. Thermal Science, 2011, 15(1): 81-90.

------“Tomasz Urbaumski et al. conducted relevant research on the deformation of fixed plate edges due to butt joints, and analyzed the technical and structural parameters to evaluate the deformation shape [15].”

[15] Tomasz Urbański, Andrzej Banaszek, Wojciech Jurczak. Predictions of distortion of the fixed plate edge of the basis of designed experiment[J]. Polish Maritime Research, 2020, 27(1): 134-142.

Comment 2: There is a lack in the work , at each of the four conditions of operation of the frame of a motor vehicle to provide a method of determining the maximum loads of the analyzed frame. (Load model of the analyzed frame structure in FEM with specific load values) This is very important for the stage of defining the finite element analysis process and possible validation of the obtained numerical simulations. Please supplement in the text of the article for better understanding by the reader of the conditions under which the analysis of the vehicle frame work was carried out and whether this is covered by the data from the measurements carried out on the actual car structure.

Response: Thanks for your comments. We have read your suggestion carefully and we would like to explain it. The frame in this article is a real pump truck product, we have a lot of design and calculation, the relevant data will be fed back to the production department. The production department will sell the products after production and communicate with customers, and get a high degree of customer recognition of the products. The simulation of this paper is based on the analysis and calculation of four typical cases in the process of product design in the early stage. Please understand.

Comment 3: Table 1: For the first item Beam Material Q345 (low alloy structural steel) Poisson's ratio was entered a value of 05. It seems that it should be 0.3. Mistake ?  Please comment, clarify and correct if necessary.

Response: Thanks for your comments. We have carefully checked the data in Table 1, and found that Poisson's ratio of 0.5 is the writing dislocation. In the simulation process, Poisson's ratio value was 0.2, and we have timely corrected it in Table 1 in line 113. Please check it.

------

Table 1. Frame material parameters.

Name

Materials

Density/(g·cm-3

Poisson's ratio

Elastic Modulus/GPa

Yield strength /MPa

Beam

Q345

7.85

0.2

206

345

Axles, axle supports, Support plates

40Cr

7.85

0.3

211

785

Tires

Rubber

1.2

0.47

7.8×10-3

-

Other parts

Q235

7.85

0.3

210

235

Comment 4: The inscriptions on the diagrams of Fig. 8, 10, 12, 14, 16, 18 , 20 are not enough legible.. Please correct to make them more legible.

Response: Thanks for your comments. We have carefully changed the unclear words in Fig. 6, 8, 10, 12, 14, 16, 18, and 20. Please check it.

------

(a) Axle displacement curve                            (b) Crossbeam displacement curve

(c) Longitudinal beam displacement curve

Figure 6. Displacement curves of main components under bending conditions.

(a) Axle stress curve                                               (b) Crossbeam stress curve

(c) Longitudinal beam stress curve

Figure 8. Stress curves of main components under bending condition.

(a) Axle displacement curve                                (b) Crossbeam displacement curve

(c) Longitudinal beam displacement curve

Figure 10. Displacement curves of main components under torsion condition.

(a) Axle stress curve                                               (b) Crossbeam stress curve

(c) Longitudinal beam stress curve

Figure 12. Stress curves of main components under torsion condition.

(a) Axle displacement curve                                (b) Crossbeam displacement curve

(c) Longitudinal beam displacement curve

Figure 14. Displacement curves of main components under turning condition.

(a) Axle stress curve                                               (b) Crossbeam stress curve

(c) Longitudinal beam stress curve

Figure 16. Stress curves of main components under turning condition.

(a) Axle displacement curve                                (b) Crossbeam displacement curve

(c) Longitudinal beam displacement curve

Figure 18. Displacement curves of main components under braking condition.

(a) Axle stress curve                                               (b) Crossbeam stress curve

(c) Longitudinal beam stress curve

Figure 20. Stress curves of main components under braking condition.

Round 2

Reviewer 1 Report

Dear Editor,

Although the authors have addressed all of the review comments and provided answers, Figures 5, 7, 9, 11, 13, 15, 17, and 19 need to be revised, and units should be added to the legends.

In conclusion, after the suggested minor revisions are made, the reviewer recommends this paper for publication in Appl. Sci.

Author Response

Dear Editor:

We have restudied the valuable comments from reviewers carefully, and answer or explain these questions. The revised contents are coloured yellow in the final revised manuscript again. The point to point responds to the reviewer’s comments are listed as following again. Please check it.

Response to reviewer’s comments:

Reviewer: 1

Comment 1: Figures 5, 7, 9, 11, 13, 15, 17, and 19 need to be revised, and units should be added to the legends.

Response: Thanks for your comments. We have carefully added units to the legend in Fig. 5, 7, 9, 11, 13, 15, 17, and 19. Please check it.
